# Construction of a Diagnostic Algorithm for Diagnosis of Adult Asthma Using Machine Learning with Random Forest and XGBoost

**DOI:** 10.3390/diagnostics13193069

**Published:** 2023-09-27

**Authors:** Katsuyuki Tomita, Akira Yamasaki, Ryohei Katou, Tomoyuki Ikeuchi, Hirokazu Touge, Hiroyuki Sano, Yuji Tohda

**Affiliations:** 1Department of Respiratory Medicine, Yonago Medical Center, National Hospital Organization, Yonago 683-0006, Japan; rkato0302@gmail.com (R.K.); ike10moy@gmail.com (T.I.); toge.hirokazu.dw@mail.hosp.go.jp (H.T.); 2Division of Respiratory Medicine and Rheumatology, Department of Multidisciplinary Internal Medicine, School of Medicine, Tottori University, Yonago 683-8503, Japan; yamasaki@tottori-u.ac.jp; 3Allergy Center, Kindai University Hospital, Osakasayama 589-8511, Japan; hsano@med.kindai.ac.jp; 4Department of Respiratory and Allergorogy, Kindai University, Osakasayama 589-8511, Japan; tohda@med.kindai.ac.jp

**Keywords:** adult asthma, artificial intelligence, diagnostic assistant, machine learning, random forest, XGBoost

## Abstract

An evidence-based diagnostic algorithm for adult asthma is necessary for effective treatment and management. We present a diagnostic algorithm that utilizes a random forest (RF) and an optimized eXtreme Gradient Boosting (XGBoost) classifier to diagnose adult asthma as an auxiliary tool. Data were gathered from the medical records of 566 adult outpatients who visited Kindai University Hospital with complaints of nonspecific respiratory symptoms. Specialists made a thorough diagnosis of asthma based on symptoms, physical indicators, and objective testing, including airway hyperresponsiveness. We used two decision-tree classifiers to identify the diagnostic algorithms: RF and XGBoost. Bayesian optimization was used to optimize the hyperparameters of RF and XGBoost. Accuracy and area under the curve (AUC) were used as evaluation metrics. The XGBoost classifier outperformed the RF classifier with an accuracy of 81% and an AUC of 85%. A combination of symptom–physical signs and lung function tests was successfully used to construct a diagnostic algorithm on importance features for diagnosing adult asthma. These results indicate that the proposed model can be reliably used to construct diagnostic algorithms with selected features from objective tests in different settings.

## 1. Introduction

Asthma is characterized by a history of symptoms of airflow obstruction such as wheezing, shortness of breath, chest tightness, and coughing that vary substantially in magnitude, spontaneously or with treatment [1]. Airway hyperresponsiveness (AHR) or objective evidence of airflow obstruction with partial reversibility is used to make the diagnosis of asthma. However, lung function tests for objective evidence of airflow obstruction are less often performed for diagnosing adult asthma in the real world [2,3]. A systematic review and meta-analysis showed that the bronchodilator response (BR), which was performed as objective evidence of airflow reversibility, has a limited sensitivity of 38.9% and specificity of 94.6% [4]. No “gold” reference standard is available for confirming or overturning the diagnosis [5].

The misdiagnosis of asthma, including overdiagnosis and underdiagnosis, is partly due to the lack of a standard approach for diagnosing adult asthma. Routine objective lung function tests likely reduce overdiagnosis before any treatment is commenced [6]. Poor disease outcomes and unwanted side effects without any clinical benefit are linked to improper use of asthma treatment [7,8,9,10].

Diagnostic prediction tools should provide realistic probabilities of a diagnosis and practical guidance on available tests in order to be adopted by general practitioners (GPs) and used in everyday practice [11]. Four national and international guidelines have been developed as evidence-based diagnostic algorithms or flow charts to diagnose adult asthma [1,5,12,13]; however, these provide conflicting advice for GPs. The National Institute for Health and Care Excellence suggests diagnosing using objective tests on treatable traits by including a high fraction of exhaled nitric oxide (FeNO ≥ 40 ppb), which is predictive of corticosteroid responsiveness [5]. The task force established by the European Respiratory Society recommends assessing FeNO as part of the diagnostic workup for patients with suspected asthma in those whose diagnosis is not established based on first spirometry combined with bronchodilator reversibility tests [12]. In partnership with the Scottish Intercollegiate Guideline Network (BTS/SIGN), the British Thoracic Society recommends that patients have a high probability of asthma based on structured clinical assessment alone without any required objective tests [13]. On the other hand, Global Initiative for Asthma (GINA) guidelines state that objective tests assessing airway inflammation or bronchial hyperresponsiveness are not necessary for asthma diagnosis [1].

The use of machine learning (ML), a type of artificial intelligence (AI), and AI in medicine is growing. Several classical ML algorithms have been proposed, such as logistic regression analysis, support vector machine (SVM), and deep neural networks (DNN). Our earlier study was the first to show that, compared to logistic regression analysis and SVM, DNN can diagnose adult asthma at a level that is comparable to that of human specialists [14]. The internal logic of DNN needs to be explained and incomprehensible due to the complex architectures of artificial neural networks (ANN) [15]. This behavior of DNN—why it performs what it does or how it works—cannot be understood and is known as the “black-box problem.” DNN has the problem of lacking interpretability. Decision-tree-based classifiers are powerful and ultimately interpretable by enabling better output transparency [16]. Random forest (RF) and eXtreme Gradient Boosting (XGBoost) are two popular tree-structured classifiers for ML.

Auxiliary tools to diagnose adult asthma using DNN have the potential to yield misleading diagnostic results. Using decision-tree-based classifiers, this study set out to determine which features are more crucial for correctly diagnosing adult asthma. Subsequently, using two types of input features, including all full and selected features of input data for symptoms, physical signs, and FeNO, we confirm useful features for diagnosing adult asthma in various settings, assuming to diagnose adult asthma without objective evidence of airflow obstruction.

## 2. Materials and Methods

### 2.1. Patients

For this re-analysis study, 566 patients with generalized respiratory symptoms were eligible. The patients were thoroughly reviewed in our prior paper [17]. In summary, 566 cases involved 367 positive cases of asthma and 199 negative cases. The cases had a median age of 52 (18 to 88) years and included 345 women and 221 males. The breakdown of the asthmatic patients, comprehensively diagnosed by experts based on pertinent symptom history and objective tests, revealed a positive finding for reversibility with a bronchodilator response (BR) of 15%, defined as a 200 mL and a 12% increase in forced expiratory volume in one second (FEV_1_), and a positive test for airway hyperresponsiveness (AHR) of 97%. A positive methacholine-induced AHR test defines 5 mg/mL as below the provocative methacholine dose, producing a 20% reduction in the FEV_1_ (PC_20_).

### 2.2. Features of Variables for Inputs

All patient feature records were divided into four components: symptom–physical signs, blood tests, lung function measurements, and airway inflammatory tests.

Symptom–physical signs (eight features): age (actual variable), sex (dichotomous variable) (female = 0, male = 1), history of wheezing, diurnal variation of symptoms, repeated symptoms, history of allergy diseases, family history of allergy diseases, and current wheezing in auscultation as dichotomous variables (yes = 1, no = 0). Blood tests (four features): the number of peripheral eosinophils, the number of peripheral basophils, total IgE (actual variables), and a positive air-borne specific IgE (dichotomous variable) (yes = 1, no = 0). Lung function measurements (five features): percent predicted FEV_1_ (%predicted FEV_1_), the airflow rate at 50% vital capacity (V_50_), V_25_, V_50_/V_25_, and increased volume in FEV_1_ after administration of bronchodilator drug (BD) (actual variables). Airway inflammatory test (one feature): FeNO level as an actual variable.

### 2.3. Model Construction

We applied two ML classifiers—RF and XGBoost—to train a gradient-boosted decision tree with the same model representation and inference but with different training algorithms (Figure 1).

RF is a classifier consisting of a collection of tree-structured classifiers with the same distribution for all trees in the forest [18]. As base learners, RF constructs an ensemble of K decision trees (DT). Each DT individually predicts the output, and the predictions are then averaged to produce the outcome (Equation (1)):(1)$$\hat{T}(x)=\frac{1}{k}\sum_{k=1}^{K}\hat{T}k(x)$$
where x represents the input, and $\hat{T}$ k(x) represents the estimation made by the kth tree [19].

XGBoost is an ensemble tree method that uses gradient descent architecture to boost weak learners [20]. The loss function is given a regularization term by XGBoost, which smooths out the learned weights and prevents overfitting. The output of XGBoost can be calculated as follows:(2)$$ŷ=\frac{1}{k}\Sigma_{k=1}^{K}fk\left(x\right),fk\epsilon Г$$
where fk represents the output of the kth tree, x is the input vector, Γ denotes the function space containing all possible regression trees, and ŷ is the projected output [21]. Equation (3) displays the objective function of XGBoost:*Obj*(*θ*) = *L*(*θ*)+*Ω*(*θ*) (3) where *L*(*θ*) is the loss function that calculates the difference between the target value and the predicted value, and *Ω*(*θ*) is the regularization function that manages the model’s complexity and guards against overfitting [22].

The RF and XGBoost algorithms were obtained from Python’s scikit-learn function.

We prepared two types of input data to construct a diagnostic algorithm for adult asthma diagnosis. In the first model, all full features of input data consisted of all four components: symptom–physical signs, blood tests, lung function measurement, and airway inflammatory tests. In the second model, selected features of input data consisted of two limited components: symptom–physical signs and airway inflammatory test. During the training process of the decision-tree classifier, the total dataset was divided into three datasets: training, validation, and test sets.

### 2.4. Hyperparameter Optimization Using Optuna

Bayesian optimization effectively optimizes objective functions globally and builds a smooth model [23,24]. The training was performed using Optuna [25], which is available as a library for Python. To avoid overfitting the model, we used Optuna, which hyperparameters for RF and XGBoost classifiers can tune. In the process of the RF classifier, Gini impurity for the supported criterion and the maximum depth of the tree (max_depth = 3) were fixed. The number of decision trees (n_estimators), the number of samples needed to split an internal node (min_samples_split), and the number of features to take into account when determining the best split (max_features) were adjusted. In the process of the XGBoost classifier, some parameters were fixed on Gini impurity for the supported criterion, maximum depth of the tree (max_depth = 3), early stopping at 50, and learning rate. The other parameters were tuned using Optuna on L1 regularization (reg_alpha), L2 regularization (reg_lambda), a fixed threshold of gain improvement to keep a split (gamma), minimum sum of Hessians needed to keep a child node (min_child_weight), subsample rows of training data prior to fitting a new estimator (subsample), and fraction of features for subsampling at different distinctions in the tree building process (colsample_bytree).

### 2.5. Performance Evaluation of Feature Importance and Metrics

Internal estimates were used to evaluate feature importance and provide global explanations for the prediction process of each selected parameter to measure feature importance using SHapley Additive exPlanations (SHAP) [26]. The global average and model output can be fully explained using SHAP, which can also take into account single and multiple correlations for linear and nonlinear interactions. Game theory provides a strong theoretical framework for SHAP. To do this, SHAP gives each variable a score that represents its relative importance in determining the result. The following is how SHAP defines a model’s output:$$f\left(x\right)=g\left(z^{’}\right)=\Phi 0+\Sigma_{i-1}^{M}\Phi iz_{i}^{’}$$
where g(z′) denotes the SHAP explanation model, ϕi represents the Shapley value, and ziϵ{0, 1} is a binary variable [27].

Decision trees were visualized using the Python library dtreeviz. The k-fold cross-validation was used to validate the calculated methods. The original dataset was split into k number of folds. The remaining k − 1 folds served as the training set, and each fold was utilized once as a validation set. Five evaluation metrics, namely precision, recall, F1 score, accuracy, and AUC (area under the receiver operating characteristic curve), were used to assess model performance. Among the total test set, accuracy was the proportion of patients that were correctly predicted. ROC (receiver operating characteristic) curves were generated by plotting sensitivity versus specificity at different classification thresholds.

## 3. Results

### 3.1. Learning Curve and Hyperparameters

The optimized hyperparameters for the best performance of the RF and XGBoost classifier on the first and second models of input datasets are shown in Table 1. These conditions were selected based on accuracy and overfitting. Overfitting is a problem with sophisticated nonlinear learning algorithms such as gradient boosting.

We used learning curves in ML to optimize the internal parameters of algorithms. Figure 2 displays the training and validation accuracy scores for the proposed XGBoost classifier on the first model input dataset. The training and cross-validation scores were approximately equal algorithms using full components.

### 3.2. Performance of Algorithms

Table 2 summarizes the performance of the classifiers using k-fold (k = 10) cross-validation in prediction algorithms for diagnosis of adult asthma. The performance of the XGBoost classifier using the first model of input datasets was the highest compared with those of all other combinations of classifiers and dataset models, with a precision of 0.80, a recall of 0.81, an F1 score of 0.81, an accuracy of 0.81, and an AUC of 0.81. Figure 3 more unequivocally illustrates the AUC of these models.

### 3.3. Feature Importance and Visualization of Diagnostic Algorithms

The Shapley value, used as an index of feature importance, is the average of all marginal contributions to all possible coalitions. Figure 4 depicts a SHAP beeswarm plot, where dots stand in for individual cases and are color-coded according to the variable’s value on the *y*-axis and their related Shapley value on the *x*-axis. We acquire information on the variable’s value, computed by the XGBoost classifier on the first and second models of the input datasets, as well as the amount and directionality of contribution to prediction to diagnose adult asthma. When all features were entered into the first model, %V_50_ was identified as the most crucial feature, followed by %predicted FEV_1_ and total serum IgE value, with a combined weighted score of more than 30% (Figure 4A). A diagnostic algorithm was constructed based on feature importance patterns. Figure 5A visualizes the diagnostic algorithm using the XGBoost classifier on an all-features model. The decision tree starts with %predicted FEV_1_ as the root node, the topmost node in a tree data structure. Then, the root node is split into either the number of peripheral eosinophils or %V_50_ as the next decision node.

As selected features of input data consist of two limited components, such as symptom–physical signs and airway inflammatory tests used to construct the diagnostic algorithm, diurnal variation of symptoms, repeated symptoms, and FeNO levels were chosen as more important features (Figure 4B). The diagnostic algorithm was visualized using the XGBoost classifier on the part of the feature model in Figure 5B. The decision tree starts with repeated symptoms as the root node and then splits into FeNO levels.

## 4. Discussion

To promote the clinical use of deep learning, studies must address the major hurdle of model interpretability through the “black-box problem.” This study demonstrated how features can be useful in unraveling explanations that have shown improvement in diagnosing adult asthma using a decision-tree-based classifier. First, compared with the two classifiers, the XGBoost classifier was a better model than the RF classifier regarding good performance accuracy for diagnosing adult asthma. Second, two different diagnostic algorithms were automatically constructed using a decision-tree-based classifier of the XGBoost classifier on different combinations, including all full and selected features consisting of symptom–physical signs, lung function measurements, and FeNO as inputs.

We have previously examined the effectiveness of DNN in modeling combinations of symptom–physical signs and objective tests to predict the initial diagnosis of adult asthma using ML [14]. Despite DNN being an excellent classifier for decision making, it cannot demonstrate which features are more useful for diagnosing adult asthma using ML. Decision-tree-based classifiers offer further insights into the relationship between features and targets to predict. This study investigated the capacity of decision-tree-based classifiers RF and XGBoost. Table 2 and Figure 3 indicate that the XGBoost classifier was prominent in distinguishing between asthma and non-asthma compared with the RF classifier, especially when all full features were used as inputs.

In this study, we showed feature ranking for diagnosing adult asthma using a decision-tree-based classifier. The XGBoost classifier provided interpretability as to which feature has contributed to model learning in data interpretation and model improvement. The %V_50_, %predicted FEV_1_, serum total IgE level, and age values were the top-ranking variables by the size of their impact on the prediction of adult asthma diagnosis (Figure 4A).

Shapley values can roughly evaluate the contributions of features to model learning in decision-tree-based models for classifiers. In this study, small airway dysfunction was the most important feature for diagnosing adult asthma using ML. All full features, including lung function measurements, were used to construct the diagnostic algorithm. Some reports have suggested that lung function measurements are required to accurately diagnose adult asthma [4,21,28,29]. The BTS/SIGN asthma guideline [13] advocates clinical diagnosis based on physician evaluation and promotes unbiased research to show variable airflow obstruction or AHR. Asthma is characterized by a high prevalence of small airway dysfunction, and some investigations indicate that small airway involvement is among the early clinical symptoms [30,31,32].

On the other hand, when the objective test of lung function was not used to construct the XGBoost classifier, we demonstrated that FeNO was replaced as an important feature for diagnosing adult asthma using the decision-tree-based classifier. GPs look forward to receiving an algorithm for detecting asthma in a case without airflow obstruction because the AHR test is not available in routine medical care. Our model was built on supervised learning training, where each specialist made a diagnosis of asthma based on the AHR test. However, our input data were not included in the AHR test. Additionally, GPs struggled to diagnose adult asthma in patients who presented with symptoms but did not have airflow obstruction or who refused to take an objective test of lung function test. We believe that our model will indicate whether or not these patients have asthma. Our research may help bridge the diagnostic gap between the high prevalence of AHR (97%) and the low prevalence of reversibility with BR (15%) in adult asthma.

Some studies have demonstrated that single-use FeNO contributes more towards ruling in than ruling out a diagnosis of asthma with an overall sensitivity of 0.65 and overall specificity of 0.82 in meta-analysis [4,33]. We emphasized that the combination of symptom–sign features and FeNO was accurate for diagnosing adult asthma using the XGBboost classifier. Among symptom–sign features, repeated episodes of symptoms were the more useful features in our XGB classifier using two limited components, including symptom–sign features and FeNO. Our previous report indicated that repeated episodes of symptoms were most useful for diagnosing adult asthma in logistic regression analysis of symptom–physical signs (odds ratio: 4.14; 95% CI: 2.43–7.06) [17].

We utilized the RF and XGBoost classifier, two decision-tree-based classifiers. They are ‘state-of-the-art’ classification methods using gradient boosting algorithms, including CatBoost (Category Boosting) [34] and LightGBM (Light Gradient Boosted Machine) [35] as well as XGBoost. It was reported that the superiority of these approaches depends on the dataset; XGboost sometimes performs slightly better, or LightGBM (or Catboost) does [36].

This study has some limitations. First, our study had a medium size and internal validation of analysis. A decision-tree classifier is one form of supervised ML, and it requires a supervisor’s presence for the final and correct diagnosis. Future work will require larger size and external validation through collecting patients who have been accurately diagnosed using AHR as a multiple-center study. Second, the balance of datasets affecting decision-tree classifiers is the better option to fit models. The two decision-tree classifiers that were used in this study have different training algorithms, such as algorithms built parallel in RF and sequentially in XGBoost. In this study, XGBoost was a good option due to the unbalanced data.

## 5. Conclusions

We adjusted parameters in XGBoost models and used these models to limit the misdiagnosis of asthma because DNN frequently gives misleading diagnostic results by overfitting models. Additionally, the DNN is frequently effective and accurate for diagnosing adult asthma; nevertheless, it has also been referred to as having “black-box problems” because it is unclear how the DNN “learns.” To concentrate on the interpretability of the learning process, we employed the white box of XGBoost. We propose that an optimized XGBoost model might lessen misdiagnosis. Four sets of national and international asthma guidelines advocate using diagnostic algorithms [1,5,12,13]. In this study, we established the decision-tree-based diagnostic algorithm to reveal discriminative features for the diagnosis of adult asthma. The goal of the current work was to find an efficient and usable AI model that may be utilized to diagnose adult asthma in primary care settings where GPs may choose to empirically diagnose and treat patients for asthma without objective testing of lung function. Hopefully, such a model may eventually be developed into a tailored diagnostic tool to reduce the rates of asthma misdiagnosis.

## Figures and Tables

**Figure 1 diagnostics-13-03069-f001:**
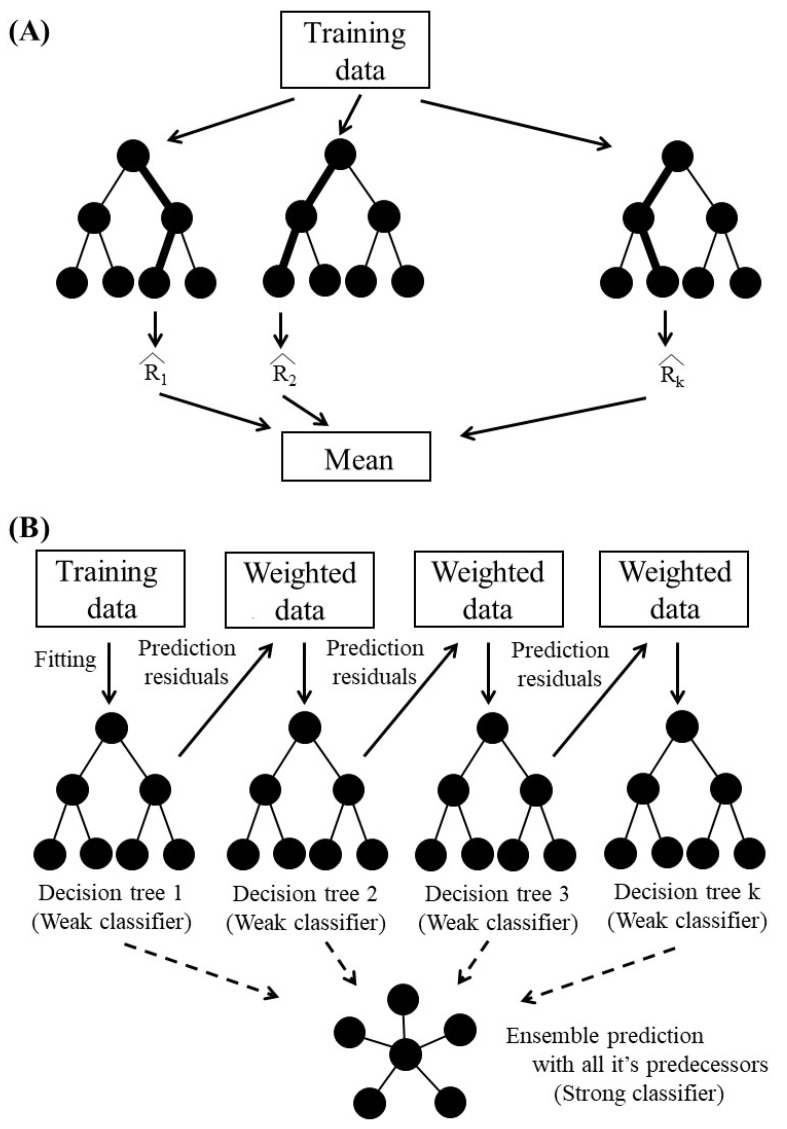
Overview of decision-tree classifiers. (**A**) Illustration of an RF classifier, which is a bagging model that trains multiple trees in parallel and determines its final output by majority voting among trees. The RF builds trees with different data and features and selects the best tree. (**B**) Illustration of an XGBoost classifier, which creates a sequential ensemble of tree models that work together to improve each other and determine its final output. It calculates an optimized tree every cycle as every new estimator is added. Each decision tree creates a series of decision rules to predict adult asthma outcomes based on input features in training data.

**Figure 2 diagnostics-13-03069-f002:**
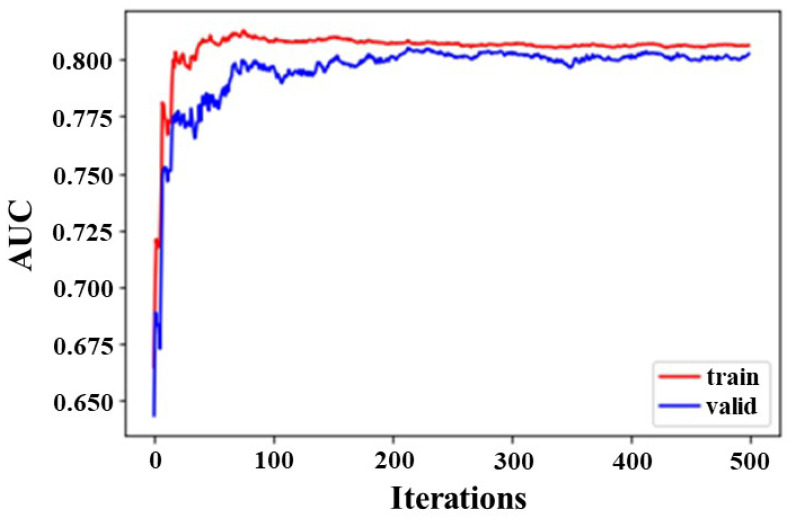
The learning curve of training and validation datasets using XGBoost classifier on the first model of input datasets using full components (18 features). The figure shows the learning curve of the training and test datasets, where the *x*-axis is the number of iterations of the algorithm (or the number of trees added to the set), and the *y*-axis is the logarithmic loss of the model. Each row shows the logarithmic loss per iteration for a given dataset. From the learning curve, the model performance (red line) on the training data set is better or has a lower loss than the model performance (blue line) on the validation data set. AUC, area under the receiver operating curve.

**Figure 3 diagnostics-13-03069-f003:**
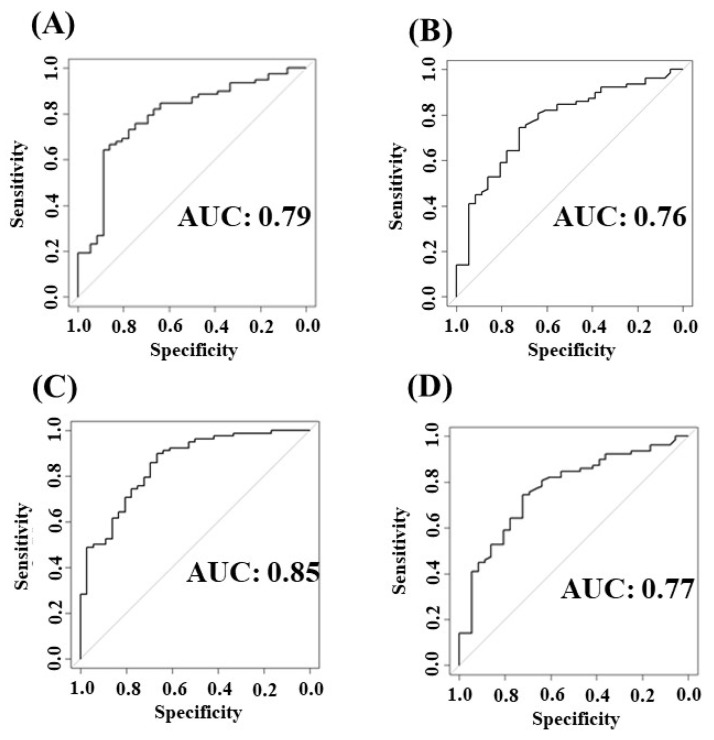
Performance of decision-tree classifiers for diagnosis of adult asthma using the ROC analysis. ROC using RF classifier on the first model of datasets (**A**), RF classifier on the second model of datasets (**B**), XGBoost classifier on the first model of datasets (**C**), and XGBoost classifier on the second model of datasets (**D**). The first model of input datasets consisted of full components, including symptom–physical signs, blood tests, lung function measurements, and airway inflammatory tests, and the second model of input datasets consisted of two components, including symptom–physical signs and airway inflammatory tests.

**Figure 4 diagnostics-13-03069-f004:**
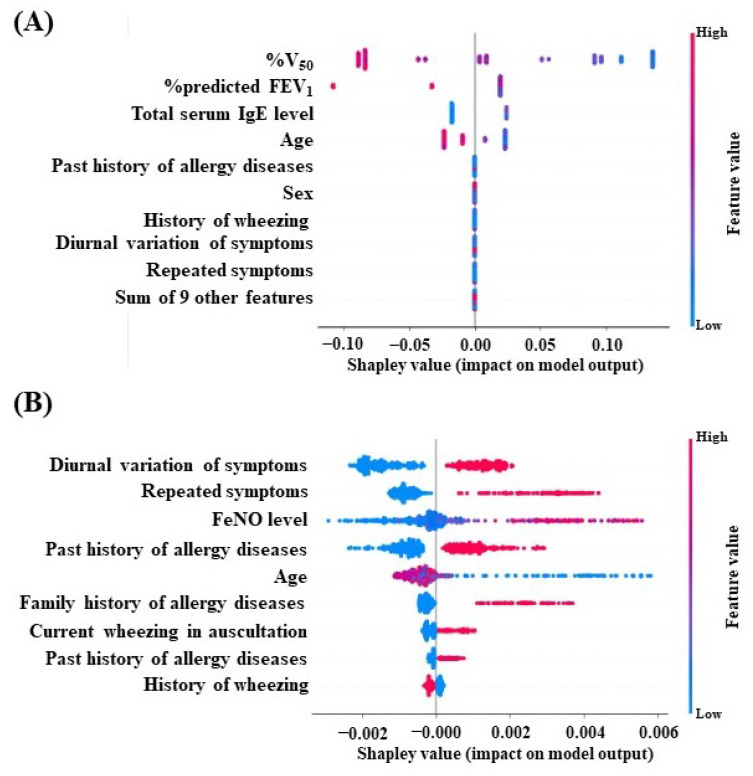
SHAP beeswarm plot demonstrating feature importance and the effects of the predictions using RF (**A**) and XGBoost classifiers on the first model of input datasets, which consisted of full components, including symptom–physical signs, blood tests, lung function measurements, and airway inflammatory test (**A**), and the second model of input data sets, which consisted of two components, including symptom–physical signs and airway inflammatory test (**B**). An adult asthma diagnosis feature’s Shapley value is represented by each point on the *x*-axis. The sum of the Shapley value magnitudes is used to order the features along the *y*-axis from top to bottom. The value of the characteristic, which ranges from low to high according to the model’s predictions, is represented by the color spectrum from blue to red. Abbreviations: FeNO, fraction of exhaled nitric oxide; %predicted FEV_1_, percent predicted FEV_1_; V_50_, airflow rate at 50% vital capacity.

**Figure 5 diagnostics-13-03069-f005:**
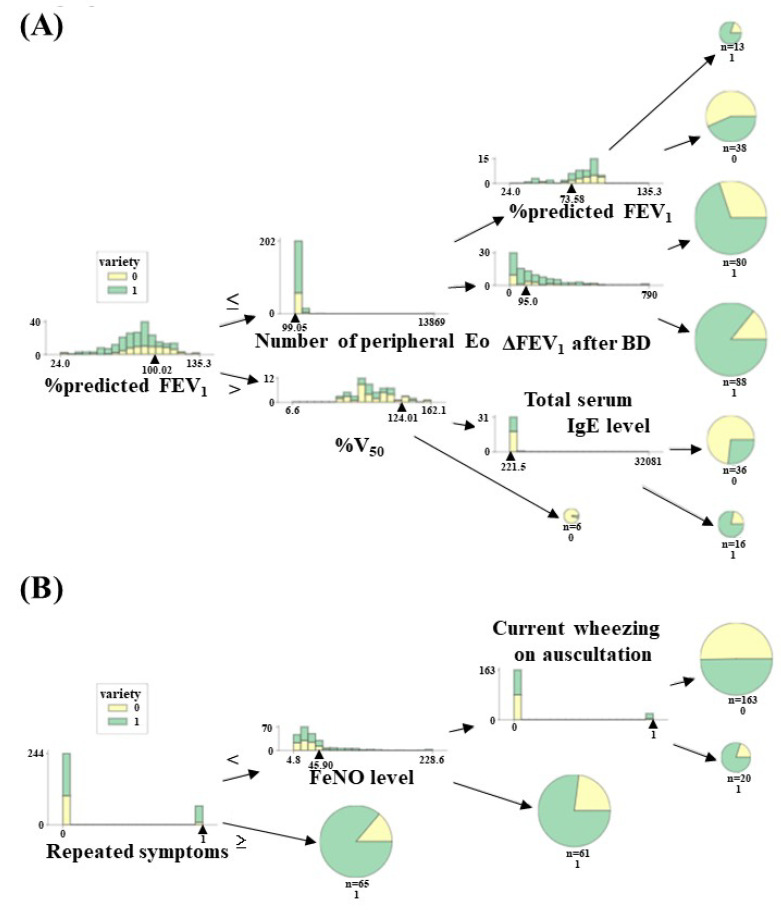
Visualization of decision tree constructed with XGBoost algorithm using the first model of input datasets, which consisted of full components, including symptom–physical signs, blood tests, lung function measurements, and airway inflammatory test (**A**), and the second model of input datasets, which consisted of two components, including symptom–physical signs and airway inflammatory test (**B**). Abbreviations: Eo, eosinophils; ΔFEV_1_ after BD, increase in volume in forced expiratory volume in one second after BD; FeNO, fraction of exhaled nitric oxide; %predicted FEV_1_, percent predicted FEV_1_; V_50_, air flow rate at 50% vital capacity.

**Table 1 diagnostics-13-03069-t001:** Optimal hyperparameter utilizing Optuna for the RF and XGBoost classifiers of the first model and the second model.

Parameters	Function	Optimal Parameter Values	
		RF *	XGBoost *
learning_rate	Learning rate in integration	NA	1 × 10^−7^–5 × 10^−8^
max_depth	Maximum tree depth	3–3	3–3
n_estimators	Number of decision trees or weak classifiers	122–346	200–358
min_samples_spit	Minimum sample weights needed to split a leaf node	26–40	NA
min_samples_leaf	Minimum number of samples required to be at a leaf node	10–3	NA
max_leaf_nodes	Total number of terminal nodes in a tree	972–292	NA
early stopping	Number of pruners of unpromising trials to stop	NA	50
min_child_weight	Minimum sum of Hessians needed to keep a child node	NA	3–2
subsample	Subsample rows of training data prior to fitting a new estimator	NA	0.5–0.5
colsample_bytree	Fraction of features for subsampling at different distinctions in tree building process	NA	0.7–0.7
reg_alpha	L1 regularization	NA	0.0047–1.138 × 10^−8^
reg_lambda	L2 regularization	NA	1.9 × 10^−7^–0.0003
gamma	Fixed threshold of gain improvement to keep a split	NA	2–2

NA, not available. * It denotes (the value of the first model–the value of the second model). In the first model, the input dataset consisted of all four feature components, including symptom–physical signs, laboratory findings, lung function tests, and airway inflammatory tests. In the second model, the input dataset consisted of two limited components: symptom–physical signs and FeNO. During the training process of decision-tree classifiers, the total dataset was divided into training, verification, and test sets.

**Table 2 diagnostics-13-03069-t002:** Performance comparison of RF and XGBoost classifiers.

Classifier	Data Set Model	Precision	Recall	F1 Score	Accuracy	AUC
RF	First model	0.66	0.75	0.72	0.76	0.79
RF	Second model	0.62	0.72	0.68	0.72	0.76
XGBoost	First model	0.80	0.81	0.81	0.81	0.85
XGBoost	Second model	0.72	0.74	0.73	0.74	0.77

AUC, the area under the receiver operating curve. In the first model, the input dataset consisted of all four feature components, including symptom–physical signs, laboratory findings, lung function tests, and airway inflammatory test. In the second model, the input dataset consisted of two limited components: symptom–physical signs and airway inflammatory test. During the training process of decision-tree classifiers, the total dataset was divided into training, verification, and test sets.

## Data Availability

The datasets used and/or analyzed during the current study are available from the corresponding author upon reasonable request.

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
