# Peer review of "Construction of a Diagnostic Algorithm for Diagnosis of Adult Asthma Using Machine Learning with Random Forest and XGBoost"

_diagnostics, 2023, doi:10.3390/diagnostics13193069_

Round 1
Reviewer 1 Report
The overall structure of the paper is appropriate. Some comments are provided to improve the quality of this manuscript. In my opinion, this manuscript should be reconsidered after major revision.
1) The literature review in the introduction section is too short. The authors should add more related works to this section.
2) The research gap in the introduction is not clear. You should clarify the research gap of previous research and highlight your contributions.
3) The description of XGBoost, SHAP, and Random Forest in section 2 is too short. Please add a subsection for each algorithm and add more details. You should add main equations of each algorithm to these subsections. Please refer to the following papers:
SHAP:
[1] “Modeling industrial hydrocyclone operational variables by SHAP-CatBoost - A ‘conscious lab’ approach,” Powder Technol., vol. 420, p. 118416, 2023, doi: 10.1016/j.powtec.2023.118416.
Random Forest:
[2] “Modeling operational cement rotary kiln variables with explainable artificial intelligence methods – a ‘conscious lab’ development,” Part. Sci. Technol., vol. 41, no. 5, pp. 715–724, 2023, doi: 10.1080/02726351.2022.2135470.
XGBoost:
4) The authors provided the parameters' value in the text. Please consider replace it with a table and list the parameters in the table.
5) No statistical tests were performed in the paper. So, how could we determine whether the results are statistically significant?
6) The English in the present paper is not of publication quality and require major improvement. Please carefully proof-read spell check to eliminate grammatical errors.
7) Please provide more discussion on the results.
8) Please add SHAP beeswarm plots to the manuscript.
9) The authors should add more evaluation metrics, such as precision, recall, and F1 score to Table 1.
10) Quality of figures is too low. Please increase the resolution of figures.
11) The authors should compare their results with state-of-the-art methods. They just compared XGBoost and Random Forest in the manuscript.
12) The methodology should be deepened.
13) Please add confusion matrix of different models.
The English in the present paper is not of publication quality and require major improvement. Please carefully proof-read spell check to eliminate grammatical errors.
Author Response
Responses to Reviewer #1
We are grateful for your interest in our work on "Construction of a diagnostic algorithm for diagnosis of adult asthma using machine learning with random forest and XGBoost". We thank you very much for your comments and useful advice. The following are our responses.
Comment 1: The literature review in the introduction section is too short. The authors should add more related works to this section.
Response 1: Thank you for your opinions. However, there has been only one report in this field to diagnose asthma using machine learning.
Comment 2: The research gap in the introduction is not clear. You should clarify the research gap of previous research and highlight your contributions.
Response 2: We add a sentence to the Introduction section as below: “Auxiliary tools to diagnose adult asthma using DNN have the potential to yield misleading diagnostic results. Using decision-tree-based classifiers, this study set out to determine which features are more crucial for correctly diagnosing adult asthma. Subsequently, using two types of input features, including all full and selected features of input data for symptoms, physical signs, and FeNO, we confirm useful features for diagnosing adult asthma in various settings, assuming to diagnose adult asthma without objective evidence of airflow obstruction”.
Commnet 3: The description of XGBoost, SHAP, and Random Forest in section 2 is too short. Please add a subsection for each algorithm and add more details. You should add main equations of each algorithm to these subsections.
Response 3: Thank you for your suggestion. According to your excellent suggestion, we add the main equations of each algorithm to these subsections in the Methods section and also additional references.
Comment 4: The authors provided the parameters' value in the text. Please consider replacing it with a table and list the parameters in the table.
Response 4: Thank you for your advice. We replace the parameters’ value in the text with this in Table 1.
Comment 5: No statistical tests were performed in the paper. So, how could we determine whether the results are statistically significant?
Response 5: We agree that statistical significance is one way of showing that our models should be taken accurately. However, due to a concept given the assumption that the null hypothesis is true, we believe that a confusion matrix could show how much better the model is than random guessing.
Comment 6: The English in the present paper is not of publication quality and require major improvement. Please carefully proof-read spell check to eliminate grammatical errors.
Response 6: Thank you for your suggestion. For extensive English revisions, we let our manuscript be proofread by the editing service.
Comment 7: Please provide more discussion on the results.
Response 7: Thank you for your advice. According to your advice, we add a sentence in the Discussion section as below: “We adjusted parameters in XGBoost models and used these models to limit the misdiagnosis of asthma because DNN frequently gives misleading diagnostic results by overfitting models. Additionally, the DNN is frequently effective and accurate for diagnosing adult asthma; nevertheless, it has also been referred to as having "black-box problems" because it is unclear how the DNN "learns." To concentrate on the interpretability of the learning process, we employed the white box of XGBoost. We propose that an optimized XGBoost model might lessen misdiagnosis”.
Comment 8: Please add SHAP beeswarm plots to the manuscript.
Response 8: Thank you for your advice. We replace the previous version of SHAEP bar plots with a new version of SHARP beeswarm plots in Figure 4.
Comment 9: The authors should add more evaluation metrics, such as precision, recall, and F1 score to Table 1.
Response 9: Thank you for your advice. According to your advice, we add some metrics such as precision, recall, and F1 score to Table 2.
Comment 10: Quality of figures is too low. Please increase the resolution of figures.
Response 10: Thank you for noticing this needed correction. We have rearranged the figures.
Comment 11: The authors should compare their results with state-of-the-art methods. They just compared XGBoost and Random Forest in the manuscript.
Response 11: Thank you for your suggestion. We add a sentence to the Discussion section as below: “We utilized the RF and XGBoost classifier, two decision-tree-based classifiers. They are ‘state-of-the-art’ classification methods using gradient boosting algorithms, including CatBoost (Category Boosting) and LightGBM (Light Gradient Boosted Machine) as well as XGBoost. It was reported that the superiority of these approaches depends on the dataset; XGboost sometimes performs slightly better, or LightGBM (or Catboost) does”.
Comment 12: The methodology should be deepened.
Response 12: Thank you for your suggestion. We add sentences regarding as main equations of each algorithm and SHAP to the Methods section.
Comment 13: Please add confusion matrix of different models.
Response 13: Thank you for your suggestion. We calculated the performance metrics such as precision, recall, and F1 score of different models, which were calculated on the confusion matrix. We add these metrics to Table 2. We don’t provide additional results of the confusion matrix.
Reviewer 2 Report
The article on the use of DL algorithms in diagnosing asthma is an exciting topic.
I have no complaints about the technical calculations, but the small amount of medical information on asthma catches my attention.
- I miss showing the characteristics of the study group of patients. This is important because the asthma diagnosis algorithm is based on these patients
- A significant study is the inclusion of the AHR test in diagnosing asthma in patients with symptoms but without obstruction in the spirometric test. The AHR test is not readily available in everyday practice. What would the calculations look like if the system did not know the information about the AHR test?
- Figure 3 should have AUC values in the pictures
Author Response
Responses to Reviewer #2
We are grateful for your interest in our work on "Construction of a diagnostic algorithm for diagnosis of adult asthma using machine learning with random forest and XGBoost". We thank you very much for your comments and useful advice. The following are our responses.
Comment 1: I miss showing the characteristics of the study group of patients. This is important because the asthma diagnosis algorithm is based on these patients
Response 1: Thank you for your comment. This study is a project involving the secondary analysis of existing data from medical records. The patients were thoroughly reviewed in our prior paper (Tomita K, et al. Prim Care Respir J 2013; 22:51–58).
Comment 2: A significant study is the inclusion of the AHR test in diagnosing asthma in patients with symptoms but without obstruction in the spirometric test. The AHR test is not readily available in everyday practice. What would the calculations look like if the system did not know the information about the AHR test?
Response 2: I agree with your opinion. As AHR test is not available in routine medical care, we physicians look forward to obtaining algorism for diagnosing asthma in a case without airflow obstruction. Our model constructed with the training in supervised learning, on which the specialist made a diagnosis of asthma based on AHR test. However, our input data were not included in the AHR test. In this study, we believe that our model will answer whether the symptomatic patient without any airflow limitation is asthma or not. According to your opinion, we add sentences to the Discussion section as below: “GPs look forward to receiving an algorithm for detecting asthma in a case without airflow obstruction because the AHR test is not available in routine medical care. Our model was built on supervised learning training, where each specialist made a diagnosis of asthma based on the AHR test. However, our input data were not included in the AHR test. Additionally, GPs struggled to diagnose adult asthma in patients who presented with symptoms but did not have airflow obstruction or who refused to take an objective test of lung function test. We believe that our model will indicate whether or not these patients have asthma.”
Comment 3: Figure 3 should have AUC values in the pictures.
Response 3: Thank you for your suggestion. We add AUC values in Figure 3.
Reviewer 3 Report
Perhaps it would have been appropriate to mention clearly what is known till now, about this subject. The article doesn’t mention this aspect, in an organized way.
As the authors mentioned, the study has his limitations. There is a medium-size and internal validation of analysis. Future work will require larger size and external validation. Otherwise, the conclusions don’t bring any real value in the day by day medical activity. The importance in the medical practice is not highlighted
This study wants to reveal an evidence-based diagnostic algorithm for adult asthma. So, it shows improvement in diagnosing adult asthma using ML. The paper presents a diagnostic algorithm that utilizes a random forest (RF) and an 15 optimized eXtreme Gradient Boosting (XGBoost) classifier to diagnose the disease. The aim is to reduce the misdiagnosis.
The paper is well written. It holds importance in the field of research and practical applicability. It addresses a topic that is insufficiently explored and it lase room for debate. The article is well organized, and the scientific research is appropriate. There are not significantly weaknesses. It follows the pattern of a study.
Author Response
Responses to Reviewer #3
We are grateful for your interest in our work on "Construction of a diagnostic algorithm for diagnosis of adult asthma using machine learning with random forest and XGBoost". We thank you very much for your comments and useful advice. The following are our responses.
Comment 1: Perhaps it would have been appropriate to mention clearly what is known till now, about this subject. The article doesn’t mention this aspect, in an organized way.
Response 1: Thank you for your suggestion. Your suggestion helps to provide an opportunity to highlight our purpose for this study. We add sentences to the Introduction section as below: “Auxiliary tools to diagnose adult asthma using DNN have the potential to yield misleading diagnostic results. Using decision-tree-based classifiers, this study set out to determine which features are more crucial for correctly diagnosing adult asthma. Subsequently, using two types of input features, including all full and selected features of input data for symptoms, physical signs, and FeNO, we confirm useful features for diagnosing adult asthma in various settings, assuming to diagnose adult asthma without objective evidence of airflow obstruction”.
Comment 2: As the authors mentioned, the study has his limitations. There is a medium-size and internal validation of analysis. Future work will require larger size and external validation. Otherwise, the conclusions don’t bring any real value in the day by day medical activity. The importance in the medical practice is not highlighted.
Response 2: Thank you for your suggestion. We add sentences to the Discussion section as below: “We adjusted parameters in XGBoost models and used these models to limit the misdiagnosis of asthma because DNN frequently gives misleading diagnostic results by overfitting models. Additionally, the DNN is frequently effective and accurate for diagnosing adult asthma; nevertheless, it has also been referred to as having "black-box problems" because it is unclear how the DNN "learns." To concentrate on the interpretability of the learning process, we employed the white box of XGBoost. We propose that an optimized XGBoost model might lessen misdiagnosis”.
Comment 3: This study wants to reveal an evidence-based diagnostic algorithm for adult asthma. So, it shows improvement in diagnosing adult asthma using ML. The paper presents a diagnostic algorithm that utilizes a random forest (RF) and a 15 optimized eXtreme Gradient Boosting (XGBoost) classifier to diagnose the disease. The aim is to reduce the misdiagnosis.
Response 3: Thank you for reading between the lines in our manuscript. Our true motive is what you have pointed out is to reduce misdiagnosis using white-box models, which focus on interpretability. As machine learning produces deceptive diagnostic results by overfitting models, we optimized parameters in machine learning and utilized these models to reduce the misdiagnosis of asthma. According to your advice, we add sentences to the Discussion section as same as Response 2.
Comment 4: The paper is well written. It holds importance in the field of research and practical applicability. It addresses a topic that is insufficiently explored and it lase room for debate. The article is well organized, and the scientific research is appropriate. There are not significantly weaknesses. It follows the pattern of a study.
Response 4: We don't deserve such a great honor. We think we need to keep going on researching this field.
Round 2
Reviewer 1 Report
The authors have addressed my comments.
No comment
Author Response
We are grateful for your interest in our study.